# Quantification of Methylisothiazolinone and Methylchloroisothiazolinone Preservatives by High-Performance Liquid Chromatography

**DOI:** 10.3390/molecules28041760

**Published:** 2023-02-13

**Authors:** Samyah Alanazi, Hajera Tabassum, Manal Abudawood, Reem Alrashoudi, May Alrashed, Yazeed A. Alsheikh, Salma Alkaff, Manal Alghamdi, Naif Alenzi

**Affiliations:** 1Department of Clinical Laboratory Sciences, College of Applied Medical Sciences, King Saud University, Riyadh 11362, Saudi Arabia; 2Department of Chemistry, College of Science and General Studies, Alfaisal University, Riyadh 11553, Saudi Arabia; 3Biological and Environmental Sciences and Engineering Division, King Abdullah University of Science and Technology (KAUST), Thuwal 23955, Saudi Arabia; 4Research and Laboratories Sector, National Drug and Cosmetic Control Laboratories (NDCCL), Saudi Food and Drug Authority, Riyadh 13513, Saudi Arabia

**Keywords:** preservatives, methylisothiazolinone, methylchloroisothiazolinone, HPLC, baby wipes

## Abstract

Isothiazolinone preservatives (methylisothiazolinone (MIT) and methylchloroisothiazolinone (CMIT) are commonly used in cosmetics, industrial and household products. However, these isothiazolinone derivatives are known to cause allergic contact dermatitis. Hence, a sensitive, accurate, and reliable method for the detection of these compounds is thus warranted. The study aims to analyze concentrations of MIT and CMIT by high performance liquid chromatography. The analytical method used for quantification of MIT and CMIT in cosmetic products (leave-on-baby wet wipes) complies with the validation acceptance criteria (international standards ISO 5725, EU25 European Union for cosmetic regulations). MIT and CMIT were extracted and analyzed in leave-on baby wet-wipes collected from different stores in Riyadh city. Extraction was performed by ultrasonication of the samples, solid-phase extraction, and liquid-liquid extraction. Ten (10) µL of the sample was injected into the HPLC system and samples were analyzed with a mixture of acetic acid and methanol (80:20 *v*/*v*) in an isocratic mode. The flow rate was maintained at 1 mL/min. UV detection was performed at 274 nm. The results demonstrated recoveries between 90 and 106%, measurement uncertainty of C +/− 0.4% for methylisothiazolinone and C +/− 0.03% for methylchloroisothiazolinone, repeatability limit (r = 0.2%) and intermediate precision limit; R = 2% and R^2^ of 0.9996.

## 1. Introduction

Cosmetic products are essential elements in day-to-day life of human beings. In 2019, the European consumption of cosmetics reached 79.84 billion euros, with an increase of about 6% since 2012, and the numbers are expected to increase worldwide [1]. To meet the high and increasing demand for cosmetics, regulatory commissions are engrossed in controlling the safety and quality of products, more precisely in rendering safety against toxic ingredients and biological or microbial contamination. The nature of ingredients and their amounts incorporated is a major health concern in cosmetic manufacturing. Water, emulsifiers, preservatives, thickeners, moisturizers, colorants, and fragrances are the primary ingredients added to these products. To note, some of these ingredients can cause adverse reactions to consumers if used in high amounts or inappropriately. Regarding regulation, cosmetic manufacturing is well monitored in Saudi Arabia, with the quality and safety parameters of the factories being examined on a regular basis [2].

Cosmetic products are at risk of microbial contamination as they constitute an ideal environment for microbial growth owing to the presence of moisture from water, and organic/inorganic solvent added in the preparation of these products. In view of the applicability of these products for prolonged usage, effective preservatives are added that should ensure consumer safety and the good quality of the product. Of the two-natural and synthetic preservatives, synthetic preservatives are far more effective than natural preservatives owing to high efficiency at low concentrations, and antimicrobial activity against a broad spectrum of microbes. Currently, a combination of two or more synthetic preservatives has emerged as a promising antimicrobial against a broad spectrum of microbes. This fusion technique eliminates the possibility for the microbes to gain resistance and decreases the toxicity risk [3]. However, synthetic preservatives may cause undesirable side effects to consumers such as skin irritation, and unfavorable estrogenic activity [4,5]. Consequently, ingredients in cosmetic products must be carefully studied and regulations must be set by the appropriate commissions. The addition of synthetic preservatives in cosmetic products is crucial for maintaining the integrity, safety, and quality of the product. The most common preservatives used in cosmetic products are classified based on chemical composition and include aldehyde and formaldehyde releasers, heavy metal derivatives, alcohols and phenols, nitrogen composites, isothiazolinones, organic acids, inorganic compounds, quaternary ammonium compounds (QAC), and biguanides.

Isothiazolinone derivatives, particularly methylisothiazolinone (MIT) and methylchloroisothiazolinone (CMIT), are widely used in the formulation of cosmetic products for both rinse-off products (shampoo and conditioner) and leave-on products (cream, sunscreen, wet tissues), because this type of preservative manifests a great broad-spectrum antimicrobial activity at low concentration. In cosmetic manufacturing, CMIT is marketed as Kathon™ CG (trade name) in a fixed ratio of 3:1 of CMIT and MIT, respectively. Kathon™ CG is incorporated into many cosmetic products *viz.* shampoos, conditioners, hand soaps including wet wipes. Though useful, these compounds have adverse effects on the skin and eyes [6,7,8,9,10,11,12,13]. Isothiazolinones are known to cause dermal, oral, and inhalation toxicity [6,7,14]. Interference of normal development of some biological systems on exposure to isothiazolinones has also been reported [15]. In addition to the efficient qualities MIT possesses, it causes complications ranging from simple allergic reaction to neurotoxicity. In vitro experiments prove that neurons exposed to MIT will undergo pathways leading to cell death. Case reports to MIT allergy are continuously increasing. The European Union (EU) witnessed a 4.1% increase in allergic cases from 2010 to 2013. In 2013, the American Contact Dermatitis Society termed MIT as *allergen* of the year [16]. Occupational workers are more prone to be affected as they are exposed to MIT in their daily life [11]. With each country following different cosmetic regulation laws worldwide, the Saudi Food and Drug Authority (SFDA) in Saudi Arabia is responsible for safety, effectiveness, and security of foods and drugs for humans and animals; and safety of complementary chemical and biological substances, cosmetics, and pesticides; in addition to safety and accuracy of electronic devices and its impact on public health. Similarly to other countries, Saudi Arabia adopts cosmetic regulations set by the EU25 (European Environment Agency). As per the regulations, Kathon™ CG and MIT are allowed in rinse-off products only with a maximum concentration of 15 ppm. Whereas in leave-on products, Kathon™ CG and MIT are banned for use. This necessitates to have a reliable and valid analytical method for the identification and determination of the MIT and CMIT in these products.

Numerous studies have reported methods for the determination of MIT and CMIT. A solid-phase extraction technique was employed using a CHROMABOND^®^ HR-X polypropylene extraction column to extract derivatives of isothiazolinones including MIT. A NUCLEOSHELL^®^ PFP Pentafluorophenylpropyl-modified HPLC column was used for chromatographic detection [17]. Other methods include dissolving samples in the appropriate solvents under ultrasonic waves, and an RP-C18 column with different specifications were used for chromatographic detection [18]. A more complex extraction method involving a three-phase hollow-fiber liquid-phase microextraction [19] and a matrix solid-phase dispersion [20], both of which were followed by a chromatographic detection using a RP-C18 column, have been reported. More recently, mass spectrometry was reported in quantification of these products [20,21]. Despite the advantages of the aforementioned studies, the method used in these studies is sophisticated and costly. On the other hand, the method reported in the current study is comparatively simple, easy, low cost, and easy to implement for routine control. The rationale behind the study was to standardize a method for detecting these preservatives utilizing materials, and the limited resources available at the Saudi Food and Drug Authority (SFDA).

In view of the adverse effects of methylchloroisothiazolinone on human health and strict government regulations of the permissible levels of MIT, the present study was undertaken to evaluate levels of methylisothiazolinone (MIT) and methylchloroisothiazolinone (CMIT) by high-performance liquid chromatography (HPLC) method.

## 2. Results and Discussion

A total of eight samples of wet wipes were tested, six of those were positive for CMIT and three were positive for MIT. Samples are indicated only with their initials for privacy. Figure 1 represents the chromatogram of all the samples. The samples were run in triplicates and mean concentrations of MIT and CMIT were determined and shown in Table 1 and Table 2.

Chromatograms of MIT and CMIT with an elution time of 2.5 and 5.0 min, respectively are represented in Figure 2.

Figure 3 depicts the calibration curves of MIT and CMIT in samples, with unknown concentrations determined from the slopes. The curves had an R-squared value of 0.9996, indicating that reliable calculations of the samples MIT and CMIT content.

### 2.1. Results on Method Validation

The method was validated for both MIT and CMIT based on standards parameters-linearity, precision, bias, working range, and measurement uncertainty (Table 3).

### 2.2. Linearity

Analytical method linearity is defined as the ability of the method to obtain test results that are directly proportional to the analyte concentration, within a specific range. The mean peak area obtained from the chromatograms was plotted against corresponding concentrations to obtain the calibration graph. Linearity was determined by three injections of standard solution for seven levels of calibration with four different replicates (Figure 4). The results of linearity study (Figure 4) produced a linear relationship over the concentration range of 0.15 to 5.8 ppm for MIT and 0.44 to 17.43 ppm for CMIT. The acceptance criteria were if Fobs ≥ F1-α the assumption of the non-validity of the linear dynamic range is accepted (with a risk of α error of 5%). The results of linearity indicate Fobs (a value of 0.00) was ˂F _95%_ (2.68) and hence the assumption of the validity of the linear dynamic range is accepted.

### 2.3. Precision

The precision of the method is defined as “the closeness of agreement between a series of measurements.” The acceptable recovery range is in between 95 and 105% and the obtained recovery was in the accepted range (90 and 106%).

The working range is the interval over which the method provides results with an acceptable uncertainty. The working range was defined from 0.15 to 5.8 ppm for MIT and 0.44 to 17.43 ppm for CMIT with an acceptable measurement uncertainty of C +/− 0.4% for MIT and C +/− 0.03% for CMIT. Based on these results the method is proved to be the valid method for determination of MIT and CMIT in cosmetic product (wet wipes).

The result obtained in the present study is in compliance with previous studies. Similar HPLC based detection was reported by Baranowska et al., 2013 [22]. More recently, Le Lee Huong Hoa et al., 2019 have reported identical method of HPLC for quantification of MIT and CMIT in shampoos [23]. The protocol is the reported study varied with the current investigation on the nature of mobile phases and type of column used in HPLC. Nevertheless, the recovery obtained in the two studies was similar and validation in accordance with EU guidelines.

The European regulations are an international model for cosmetic regulation since they are comprehensible and application is attainable. The first regulations with reference to cosmetic regulation were published in 1976 as a consequence of several catastrophic cases of injury and death following exposure to several toxic chemicals incorporated in baby products and adult cosmetic products. These chemicals include organochlorine compounds which are widely used as pesticides, and are added as disinfectant [24]. The effects of these compounds on humans include cardiovascular disorders [25], neuromuscular disorders [26], neurotoxicity, and cancer. The regulations of 1976 were improved in 2009 and published in 2013, which include the latest technological advancement, namely, the implementation of nanomaterials [24]. Regulations concerning the use of Kathon™ CG and MIT in cosmetic products are strict. In 2009, a maximum of 15 ppm of Kathon™ CG and a maximum of 100 ppm of MIT in all products were accepted by the European parliament and the council of the European Union [1]. By 2016, the use of Kathon™ CG and MIT were banned in leave-on products and allowed for use only in rinse-off products following the same concentrations set in 2009 [17,27]. In 2017, the maximum allowed concentration for MIT use was decreased to 15 ppm in rinse-off products [18]. Since then, the regulations have not changed.

According to the Saudi regulations and EC, wet wipes are considered a leave-on products. As indicated in previous sections, MIT in leave-on products is strictly prohibited. Unfortunately, MIT and CMIT were detected in wet wipes in concentrations ranging between ~0.1 ppm to ~3 ppm, the calculated concentrations face very low MoE (margin of error). Although the results presented are convincing, there are some shortcomings in the data obtained which may offer some bias as the samples tested were not randomly chosen and the sample size was small.

## 3. Methodology

### 3.1. Materials

A total of 8 samples of wet wipes of different brands were obtained from different stores in Riyadh. Distilled and deionized water was used in all experiments. All chemicals used in the experiments were of HPLC grade. Methanol was obtained from Merck (Rahway, NJ, USA), and acetic acid was obtained from Riedel-de Haën (Seelze, Germany). For reference materials, pure materials of methylisothiazolinone (MIT) and methylchloroisothiazolinone (CMIT), HPLC grade were obtained from Merck (Darmstadt, Germany). The analytical standard Kathon^TM^ CG, a mix standard of methylisothiazolinone (MIT) and methylchloroisothiazolinone (CMIT), was purchased from SIGMA-ALDRICH (St. Louis, MO, USA). The diluent used was a mixture of 0.4% solution of acetic acid (CH_3_COOH) and methanol (CH_3_OH). The mobile phase consisted of 80% of the acetic acid solution (0.4% solution) and 20% methanol. The column used for separation was Thermo Scientific™ (Waltham, MA, USA) 250-4.6 Hypersil™ BDS (Base-Deactivated Silica) RP-C18 (5 µm) column. A SHIMADZU (Kyoto, Japan) Prominence-i series High-Performance Liquid Chromatography (HPLC) system was employed which is equipped with an autosampler, degasser, and quaternary solvent delivery unit and a PDA detector.

### 3.2. Methods

#### 3.2.1. Sample Preparation

One (1.0) g of each sample was weighed in a 50 mL beaker and soaked in 20 mL of a mixture of methanol and diluted acetic acid for about 24 h. The mixture consisted of 0.4% acetic acid and pure methanol in a ratio of 4:1. The acetic acid is incorporated in both the diluent and mobile phase to maintain a steady pH of ~3. After 24 h, beakers with samples were placed under ultrasonic waves for 90 min in ambient conditions under a Hwashin Technology Co. Powersonic 405 ultrasonic cleaner (Seoul, Republic of Korea) which is a crucial step for the extraction of the targeted analyte. The solutions were transferred to 50 mL conical centrifugation tubes and the pieces of wipes were squeezed to retrieve the maximum possible volume of the solvent. The solutions were centrifuged for 40 min at 4000 rpm under Hettich Universal 320 centrifuge (Kirchlengern, Germany). The final samples were transferred into HPLC vials after filtering the supernatant with 0.45 µm filters.

#### 3.2.2. Optimization

In the search for the most suitable extraction technique and most suitable HPLC conditions, several parameters were adapted.

##### Diluent Variations

Methanol was used in different ratios with water; 50% and 80%. Acetonitrile was also used in different ratios; 50% and 80%. Hexane and water were used separately as well (50% and 80%). A mixture of methanol, acetonitrile, and 0.4% acetic acid with the ratio 1:1:8, respectively, was used. A few drops of 1M sodium chloride were also added to some samples to reduce foam production.

##### Extraction Techniques

Solid-phase Extraction (SPE): A total of 8 columns were tested; IST ISOLUTEP^®^ ENV+, IST ISOLUTE^®^ SI, IST ISOLUTE^®^ NH2, IST EVOLUTE ABN 50, IST EVOLUTE CX 50, IST ISOLUTE MULTIMODE, VARIAN Bond Elut C18, and Agilent SAMPL10 Silica (Agilent Technologies, Santa Clara, CA, USA). The main phases of solid-phase extraction are conditioning, sample loading, washing, and eluting. Methanol was used for conditioning, water with 0.1% formic acid was used for washing, and a mixture of 4:1; acetonitrile and methanol, respectively, were used for the elution. In another trial with the IST EVOLUTE ABN 50 column, a 0.01 M sodium phosphate buffer solution was used for conditioning and the elution. The flow rate was controlled manually and was kept at low rates (1 drop/5–7 s) to maximize the interaction of the solutions with the column. After performing the solid-phase extraction, a stabilizer consisting of 20:80; 200 mM hydrochloric acid and isopropanol, respectively, was added to the samples. Following this, samples were concentrated by nitrogen evaporation.

Liquid-liquid Extraction: 50% Methanol in water as an aqueous solvent of the diluent was used and an organic layer consisting of heptane was used. Anhydrous sodium sulfate as a drying agent was also used to eliminate any aqueous parts.

##### High-Performance Liquid Chromatography

*Column*: in the search for the best separation column, two other columns were investigated. A LiChroCART 250-4 LiChrospher^®^ 100 RP-18 (5 µm) and a Thermo Scientific™ 250-4.6 Hypersil™ ODS C18 Columns (5 µm) were tested.

Mobile Phase: a total of four different compositions of the mobile phase were tested as following:(i)methanol and water(ii)0.01 M sodium phosphate buffer and acetonitrile(iii)water and acetonitrile(iv)methanol, acetonitrile, and 2% acetic acid

In addition, 0.1% of formic acid was added to some compositions to maintain pH at 2.8–3.0 and to obtain sharp peaks.

#### 3.2.3. Detection by High-Performance Liquid Chromatography

The system is equipped with an autosampler, degasser, and quaternary solvent delivery unit and a PDA detector. The chromatograms were analyzed and integrated using LabSolutions software. A volume of 10 µL of each sample was injected into the HPLC system with an isocratic elution of the previously mentioned mobile phase (80% of 0.4% acetic acid and 20% methanol) flowing at 1 mL/min of flow rate. The analyte was eluted after 5 min and detected at a wavelength of 274 nm with a reverse-phase C18 column packed with base-deactivated silica.

#### 3.2.4. Method Validation

The assessment of validation results was based on performance requirements of AOAC International [28]. The method was validated in terms of linearity, precision, and measurement uncertainty.

##### Linearity

To evaluate the linearity of the method, standard solutions of MCI and MI were prepared to obtain different exact concentrations of MI ranging from 0.15 to 5.8 ppm for MIT and 0.44 to 17.43 ppm for CMIT. Three injections from each concentration were analyzed under the same conditions.

##### Precision

Precision was determined by six individual preparations of raw materials (RM) and analyzed by two different analysts at different times, with criteria of repeatability limit r = 2.8 × S_r_ and intermediate precision limit R = 2.8 × S_R_. The estimated repeatability limit was r = 0.2% and the intermediate precision limit was R = 2%. The bias was determined by three individual preparations made from RM (0.5775 ppm MIT and 1.7325 ppm for CMIT) and analyzed for 3 days.

#### 3.2.5. Statistical Analysis

Statistical analysis was performed by SPSS software. Mean, standard deviation, population standard deviation, standard error and margin of error were calculated.

## 4. Conclusions

The method employed in the current study provides a robust approach for the quantification of MIT and CMIT in wet wipes. The method is simple, promising, and cost-effective to identify MIT and CMIT in other cosmetic products too.

## Figures and Tables

**Figure 1 molecules-28-01760-f001:**
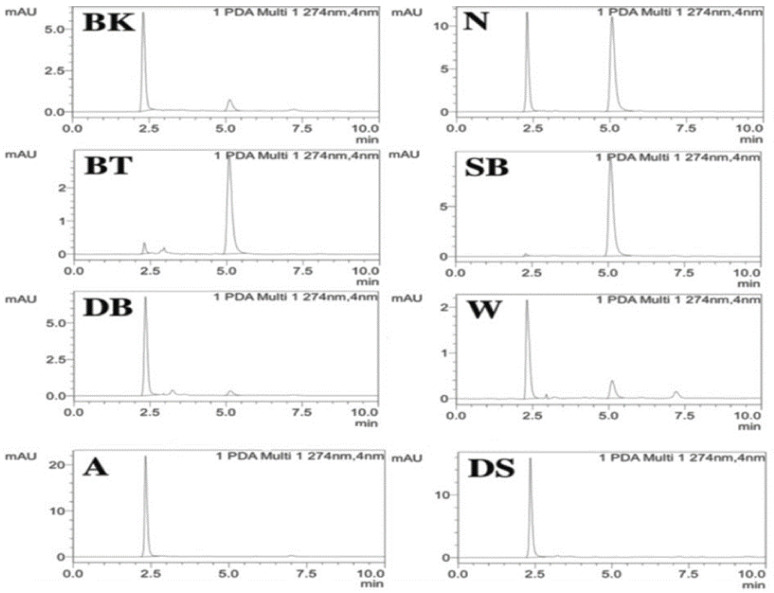
Chromatograms of wet-wipe samples.

**Figure 2 molecules-28-01760-f002:**
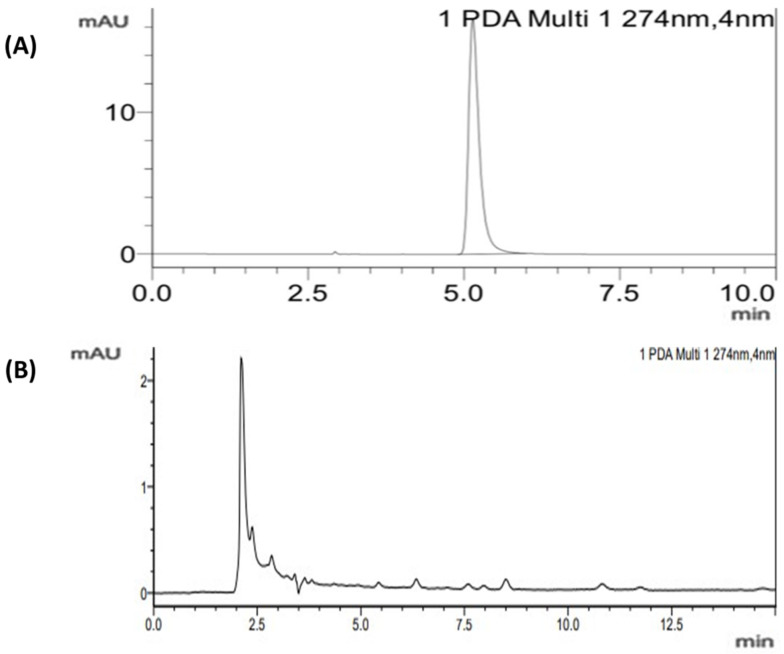
Chromatogram of (**A**) MIT and (**B**) CMIT.

**Figure 3 molecules-28-01760-f003:**
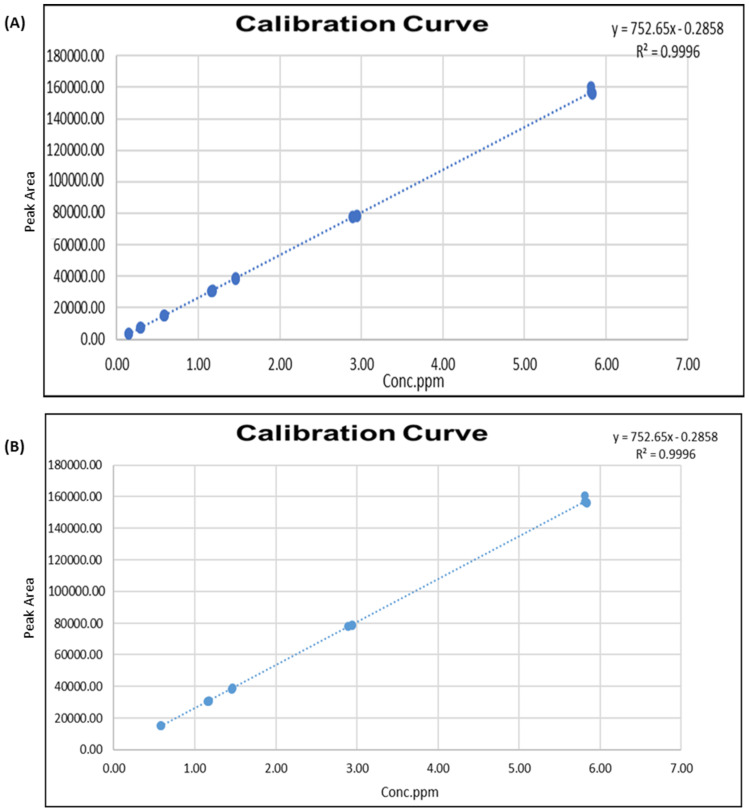
Calibration curves—(**A**) MIT and (**B**) CMIT.

**Figure 4 molecules-28-01760-f004:**
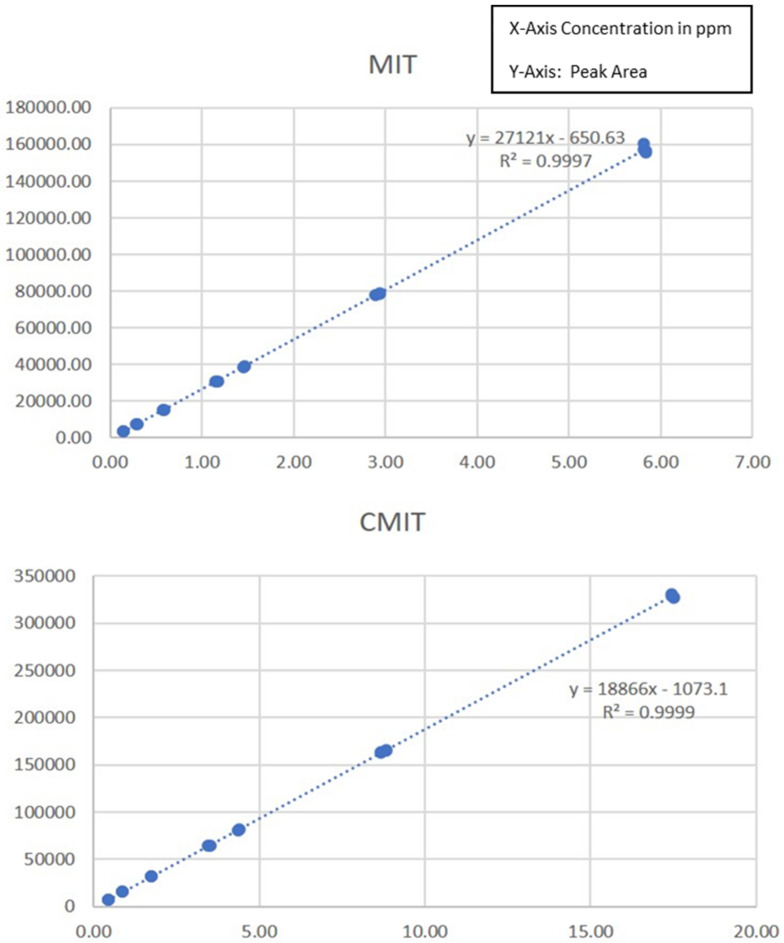
Linearity of MIT and CMIT.

**Table 1 molecules-28-01760-t001:** Concentration of MIT in wet-wipe samples.

S.No	Samples	Concentration (ppm)	Mean	Population SD	Standard Error	Sample SD
1	A	0.2147	0.2146	0.0004249	0.0002345	±0.0005
		0.2140				
		0.2150				
2	BK	0.2240	0.2231	0.0006482	0.0003742	±0.0007
		0.2226				
		0.2227				
3	BT	-	-	-	-	-
4	DB	0.1241	0.1247	0.0004063	0.0002345	±0.0005
		0.1250				
		0.1249				
5	DS	0.1295	0.1294	0.0003699	0.00125753	±0.0004
		0.1299				
		0.1289				
6	N	2.9343	2.9090	0.0411234	0.0234726	±0.05
		2.9418				
		2.8510				
7	SB	-	-	-	-	-
8	W	0.1440	0.1495	0.0039189	0.0022626	±0.004
		0.1521				
		0.1525				

SD: standard deviation.

**Table 2 molecules-28-01760-t002:** Concentration of CMIT in wet-wipe samples.

S.No	Samples	Concentration (ppm)	Mean	Population SD	Standard Error	Sample SD
1	A	-	-	-	-	-
2	BK					
3	BT	0.8275	0.8295	0.0021648	0.0012498	±0.002
		0.8285				
		0.8325				
4	DB	-	-	-	-	-
5	DS	-	-	-	-	-
6	N	3.1622	3.1652	0.0023686	0.0016298	±0.0021
		3.1680				
		3.1652				
7	SB	2.6561	2.8114	0.1710827	0.0987746	±0.2
		3.0497				
		2.7283				
8	W	-	-	-	-	-

SD: standard deviation.

**Table 3 molecules-28-01760-t003:** Results of validation studies.

Validation Parameters	Results
Linearity	Fobs (a value of 0.00) was ˂F _95%_ (2.68) so the assumption of the validity of the linear dynamic range is accepted.
Precision	Repeatability limit r and intermediate precision limit R were estimated: r = 0.2% R = 2%
Bias	A recovery between 90 and 106% was found.
Working Range	The working range is defined from 0.15 ppm to 5.8 ppm for MIT, 0.44 ppm to 17.43 ppm for CMIT with an acceptable uncertainty
Measurement Uncertainty	C +/− 0.4% for MIT C +/− 0.03% for CMIT

## Data Availability

The data sets generated during the current study are available from the corresponding author on reasonable request.

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
