# Peer review of "Quantification of Methylisothiazolinone and Methylchloroisothiazolinone Preservatives by High-Performance Liquid Chromatography"

_molecules, 2023, doi:10.3390/molecules28041760_

Round 1

Reviewer 1 Report

manuscript Development and verification of method for the detection of 2 Methylisothiazolinone and Methylchloroisothiazolinone pre-3 servatives  is reviewed and has the following limitations so not recommeded for publication in the present form 

Research has been planned for the development of method for Methylisothiazolinone and Methylchloroisothiazolinone whereas work has been conducted on the analysis and methodology do not support the title of the work and abstract information.  

Usually the method development require the pure refernce material for recovery and linearity. 

Methodology is based upon the simple analysis criteria to identify preservatives in the samples. 

No od samples are less and do not have research strength.  

Methodolgy also need improvement in respect of title and abstract

change the heading as analysis by High-performance Liquid Chromatography as detection is one part of technique by detector

Figures are very poor in resolution 

Conclusion need more tangible 

Author Response

Reviewer 1

General comments

The manuscript is revised as per the comments in the manuscript

Major comments

Research has been planned for the development of method for Methylisothiazolinone and Methylchloroisothiazolinone whereas work has been conducted on the analysis and methodology do not support the title of the work and abstract information.

The title and abstract are changed as per the methodology

Usually the method development require the pure reference material for recovery and linearity.

Pure reference material of MIT and CMIT was used. Linearity was performed using pure  MIT and CMIT (refer Figure 4)

Methodology is based upon the simple analysis criteria to identify preservatives in the samples. 

The method employed is simple, easy to determine and cost effective using available resources

No of samples are less and do not have research strength.  

We acknowledge for less number of samples used. This is added in limitations of the study. However, the results can be still validated.

Methodology also need improvement in respect of title and abstract. change the heading as analysis by High-performance Liquid Chromatography as detection is one part of technique by detector

The title and abstract are edited. The heading is changed as per the suggestions

Figures are very poor in resolution

 Unfortunately we cannot provide better resolution than the submitted figures 

Conclusion need more tangible

Conclusion is revised as suggested  

Reviewer 2 Report

Abstract Line 22, now abbreviate  methylisothiazolinone and methylchloroisothiazolinone 

Abstract. Use sentence case don't use unnecessary capital letters 

Abstract. which  validation acceptance criteria? name it  

Keywords. must be different from title page 

Line 73. in a fixed ratio of 3:1 of CMIT and crucirespectively, not understandable 

In the introduction, there is no discussion about the advantages and disadvantages of the recently reported methods. What is the problem of the previous studies that need to be solved? What is the meaning and innovation of this work? The advance of this work compared with other works should be given in detail.

No rational for present study was given 

In experimental section, sampling details with their codes were not provided 

Validation study parameters were not explained in experimental section

Which guidelines were followed for validation studies, no details were provided 

The method adopted was not compare with the reported methods for determination of MIT and CMIT

Tabulate the results and compare the results on basis of the validation parameters 

Conclusion is very weak

Author Response

Reviewer 2

General comments

The manuscript is revised as per the comments marked for the manuscript

Major comments

Abstract Line 22, now abbreviate  methylisothiazolinone and methylchloroisothiazolinone 

Changes made as suggested

Abstract. Use sentence case don't use unnecessary capital letters 

Corrections inserted

Abstract. which  validation acceptance criteria? name it  

Validation acceptance criteria from EU25 European Union for cosmetic regulations was adopted

Keywords. must be different from title page 

Corrections inserted

Line 73. in a fixed ratio of 3:1 of CMIT and crucirespectively, not understandable 

3:1 of CMIT and MIT. It was a typing error and corrected

In the introduction, there is no discussion about the advantages and disadvantages of the recently reported methods. What is the problem of the previous studies that need to be solved? What is the meaning and innovation of this work? The advance of this work compared with other works should be given in detail.

The suggested recommendations are added in the introduction section

No rational for present study was given 

The rationale of the study inserted in the manuscript in introduction.

In experimental section, sampling details with their codes were not provided 

Details included in the experimental section

Validation study parameters were not explained in experimental section

Validation study parameters are included in the method section (Table 3)

Which guidelines were followed for validation studies, no details were provided 

Guidelines from EU25 European Union for cosmetic regulations were followed

The method adopted was not compare with the reported methods for determination of MIT and CMIT

The method is compared with previous  reported studies in Result and discussion section of the manuscript

Tabulate the results and compare the results on basis of the validation parameters 

Table inserted in the manuscript as suggested.

Conclusion is very weak

Conclusion is revised as advised.

Round 2

Reviewer 1 Report

Concept of the manuscript is not clear

Author Response

Thanks for the review comment to build the manuscript up to the mark.  

Comments in round 2

Concept of the manuscript is not clear

All the comments in previous revision were reconsidered and changes inserted and highlighted. Manuscript is revised to clarify the concept.

Reviewer 2 Report

Manuscript in its current form reached the level for final acceptance for publication

Author Response

Thanks for the comment